# Vaccination in PADs

**DOI:** 10.3390/vaccines9060626

**Published:** 2021-06-09

**Authors:** Cinzia Milito, Valentina Soccodato, Giulia Collalti, Alison Lanciarotta, Ilaria Bertozzi, Marcello Rattazzi, Riccardo Scarpa, Francesco Cinetto

**Affiliations:** 1Department of Molecular Medicine, Sapienza University of Rome, 00161 Rome, Italy; valentina.soccodato@gmail.com (V.S.); giulia.collalti86@gmail.com (G.C.); 2Department of Medicine, University of Padua, 35122 Padua, Italy; alison.brokenspear@gmail.com (A.L.); ilariabertozzi2@gmail.com (I.B.); marcello.rattazzi@unipd.it (M.R.); scarpa.riccardo@gmail.com (R.S.); francesco.cinetto@unipd.it (F.C.); 3Internal Medicine I, Ca’ Foncello Hospital, 10103 Treviso, Italy

**Keywords:** immunodeficiency, antibody deficiency, immunization, vaccination recommendations

## Abstract

Primary antibody deficiencies (PADs) are the most common primary immunodeficiencies (PIDs). They can be divided into the following groups, depending on their immunological features: agammaglobulinemia; common variable immunodeficiency (CVID) isotype; hyper IgM isotype; light chain or functional deficiencies with normal B cell count; specific antibody deficiency with normal Ig concentrations and normal numbers of B cells and transient hypogammaglobulinemia of infancy. The role of vaccination in PADs is recognized as therapeutic, diagnostic and prognostic and may be used in patients with residual B-cell function to provide humoral immunity to specific infective agents. According to their content and mechanisms, vaccines are grouped as live attenuated, inactivated (conjugated, polysaccharide), mRNA or replication-deficient vector vaccines. Vaccination may be unsafe or less effective when using certain vaccines and in specific types of immunodeficiency. Inactivated vaccines can be administered in PAD patients even if they could not generate a protective response; live attenuated vaccines are not recommended in major antibody deficiencies. From December 2020, European Medicines Agency (EMA) approved vaccines against COVID-19 infection: according to ESID advises, those vaccinations are recommended in patients with PADs. No specific data are available on safety and efficacy in PAD patients.

## 1. Introduction

Primary antibody deficiencies (PADs) are the most common primary immunodeficiencies (PIDs), representing, at least, 50% of all symptomatic forms [1]. PADs can be caused by B cell-intrinsic defects or by functional impairments in other immune cell lineages, as well as innate immune cells and T cells. According to the 2019 update classification of the International Union of Immunological Societies (IUIS) Expert Committee on Primary Immunodeficiency [2], PADs can be divided into the following groups, depending on their immunological features: severe reduction in all serum immunoglobulin isotypes with profoundly decreased or absent B cells (agammaglobulinemia); severe reduction in at least two serum immunoglobulin isotypes with normal or low B cell count (CVID phenotype); severe reduction in serum IgG and IgA with normal/elevated IgM and normal B cell count (Hyper IgM isotype); light chain or functional deficiencies with generally normal numbers of B cells; specific antibody deficiency with normal Ig concentrations and normal B cell count; transient hypogammaglobulinemia of infancy with normal B cell counts.

Clinical manifestations include [3] autoimmunity and/or autoinflammation, malignancies and susceptibility to infections. The altered or absent antibody production leads to recurrent bacterial infections; most of these are caused by encapsulated bacteria, such as *Streptoccoccus pneumoniae*, *Haemophilus influenzae* and Gram-negative bacteria. These infections predominantly affect the respiratory tracts and, if untreated, lead to severe complications, such as chronic sinusitis and bronchiectasis. Less frequently, patients with PADs suffer from intestinal tract infections caused by *Giardia* spp., *Campylobacter jejuni*, *Salmonella* spp. or *Helicobacter pylori* and to bacterial cutaneous infections. Remarkably, agammaglobulinemic patients can suffer from severe, chronic enteroviral infections suggesting a major role for antibodies in preventing the dissemination of enteroviruses from the gut.

The use of IgG replacement therapy (IGRT) represents the therapeutic cornerstone of all PADs. Substitutive therapy has changed survival and reduced the incidence of pneumonia and serious infections. Igs preparations include high titers of antibodies for many vaccine-preventable viral agents except for certain viral agents such as HPV and currently circulating influenza [4]. Despite appropriate therapy, chronic manifestations such as sinusitis and bronchiectasis increase over time [5]. Patients with serum IgA levels below 7 mg/dl, severe reduction in memory B cells [6] and/or reduced or absent antibody response to polysaccharide antigens have been highlighted in patients at a high infectious risk needing additional therapeutic strategies in addition to IGRT. Therefore, administration of antibiotics and antibiotic prophylaxis might also be required [3].

The role of vaccination in PADs has long been discussed, due to impaired antibody response and IGRT. At present vaccination is recognized as a therapeutic, diagnostic and prognostic tool in this patient population.

## 2. Rationale of Vaccination in PAD

Immunocompromised patients have an increased susceptibility to vaccine-preventable infections, even if vaccination is a controversial issue in this population [1,7].

Vaccination in PADs can be used in patients with residual B-cell function to provide humoral immunity to a certain infective agent and to improve the outcomes related to vaccine-preventable disease, as demonstrated by our group studies. It has also been proven that vaccination may induce some cellular immunity [8]. In this population, vaccines may also be used to measure humoral immune function [9].

Bonilla [10] states that assessment of humoral immune function is part of the diagnostic evaluation of all patients with suspected immune deficiency; in particular, it should be evaluated for seroconversion after administering a certain vaccine, rather than the simple assessment of specific immunoglobulin isotype levels. Antibody response to vaccination can be both T-dependent and T-independent, according to the type of administered antigen. To evaluate T-dependent antibody responses, the author suggests measuring tetanus toxoid (TT) IgG (or less often diphtheria toxoids) in children older than 6 months of age (because of persistent circulation of maternal IgGs after the birth). In a healthy population a 20-to 30-fold increase in the level of TT IgG can be observed after vaccination, and the protective threshold is usually considered to be 0.15 IU/mL [10].

The gold standard to evaluate assessment of T-independent antibody responses, in clinical practice, is represented by the response to pneumococcal polysaccharide vaccines, and 23-valent pneumococcal polysaccharides vaccine (PPSV) is the most used. Antibody titers should be measured by the same laboratory before, and after 4 to 8 weeks from vaccination. The suggested IgG threshold for protection with respect to a single serotype after PPSV is 1.3 mg/mL [10]. Historically, an adequate response to pneumococcal vaccination has been defined as up to a four-fold increase in antibody titers over baseline levels depending on the different immunogenicity of each specific serotype. Age may also influence the response. Notably, higher pre-immunization antibody titers to specific serotypes are less likely to significantly increase after immunization, indicating the threshold of a two-fold increase more widely applicable independent of the pre-immunization titers [11].

Wall et al. [12] propose a different interpretation of response to PPSV. In patients who are pneumococcal protein conjugated vaccine (PCV) naive, the percentage response among all measured PPSV serotype titers resulting in the protective range should be considered. In patients who have previously received PCV, at least seven of 23 exclusive PPSV serotypes should be measured and, among all measured 23 exclusive PPSV serotypes, the percentage of those in the protective range should be considered the percentage response.

All these thresholds only take into account IgG response to immunization, while protective titers of specific anti-PPSV IgM and IgA are not defined at all.

Systematic vaccination with PCV, recommended by the Center for Disease Control and Prevention (CDC) in childhood and in at-risk-group patient schedules, may limit the use of PPSV for assessing specific polysaccharide responses, due to a “priming” effect of PCV13 in enhancing a subsequent response to PPSV [10].

In addition, patients on IGRT may present high titers of pneumococcal antibodies because preparations are rich in anti-pneumococcal IgGs. This raises the question whether to use specific IgA and IgM titers to evaluate the response. Alternative or complementary measurements of other polysaccharide responses have also been proposed, in an attempt to increase diagnostic accuracy. For example, responses to the less frequently scheduled *Salmonella typhi* pure polysaccharide vaccine (Typhim Vi) have been studied [13].

Sanchez et al. [14] demonstrated, in a multicenter study, a similar lack of response in both Typhim Vi and PPSV immunization in a group of CVID patients, suggesting that the evaluation of the specific antibody response to Typhim Vi vaccine adds clinical value to the diagnosis of anti-polysaccharide antibody production deficiency in patients with this immune defect. Another study, conducted by Guevara-Hoyer et al. [15], confirmed these data in a pediatric population previously vaccinated with the conjugate pneumococcal vaccine.

## 3. Prognostic Role of Vaccination

The capability of CVID patients to be immunized through vaccinations has not only a therapeutic and a diagnostic role, but also a prognostic role. Our study group investigated the residual ability to mount an IgM and IgA anti-polysaccharide (PS) response in CVID patients under IGRT [16]. Since a protective titer of specific anti-PPSV IgM and IgA is not precisely defined, we assumed 20 U/mL as a cut off of threshold value for protection against infections for IgM and 150 U/mL for IgA [16]. In 125 CVID patients and 20 healthy volunteers we evaluated by using an ELISA test, before, and 4 weeks after immunization with PPSV23 vaccine, the titers of IgA and IgM antibodies produced against all 23 PS serotypes. We observed that not all the 23 antigens are equally immunogenic: we noticed that PS1 induced the most robust IgM and, especially, IgA response.

On the basis of their antibody response after vaccination, patients were subdivided into three groups: group one, (91 patients, 71%), non-responders, with low specific IgA and IgM; group two, (11 patients, 9%), IgA and IgM responders; group three, (25 patients, 20%), IgM-only responders. Group 1 had a significantly lower memory B cell frequency compared to groups two and three, suggesting a correlation between anti-PPSV IgM and IgA responses and B cell subset frequencies. We also found concordance between anti-PPSV IgM and IgA levels and clinical phenotype, since we noticed a higher prevalence of pneumonia and bronchiectasis in group one compared to group two or group three [17].

We have also recently published data from a long-term observational study in the same population of CVID patients immunized by a single dose of a 23 pneumococcal polysaccharides vaccine [18]. We collected blood samples of 74 of the initial 125 CVID patients 36 (±6) months after vaccination and we quantified IgA antibodies to all 23 pneumococcal serotypes. 36 ± 6 months after the first immunization, all patients initially classified as IgA non-responders (IgA-NR) were confirmed, whereas 5 of 14 IgA responder (IgA-R) patients still presented protective IgA levels. This group of long-lasting IgA response patients showed a higher frequency of switched memory B cells in comparison to IgA-R patients who lost IgA response at the third evaluation. IgA-NR patients still showed a statistically significant increase in upper respiratory tract infections (URTI), lower respiratory tract infections (LRTI), bronchiectasis, autoimmunity and enteropathy, in comparison with IgA-R patients. All these data suggest that the quantification of PPSV IgA is useful to identify the inability to mount an IgA-mediated response against polysaccharide antigens or the inability to maintain the antibody response over time. Thus, it may represent a good clinical prognostic marker in CVID patients.

## 4. Vaccination Schedule in PAD and Recommendations

As stated before, PAD patients may show a low or absent antibody response to vaccines; however, vaccination may induce some cellular response. On the other hand, vaccination may be not safe or may be less effective when using certain vaccines and in specific types of immunodeficiency [10]. At present, since detailed information on the duration of the protection following vaccination in PAD patients is lacking, additional boosters over time may be required. Vaccines may be grossly grouped according to their content and mechanisms: live attenuated, inactivated (conjugated, polysaccharide), mRNA and replication-deficient vector vaccines.

Inactivated vaccines may be administered in primary antibody deficiencies even if they could not generate a protective response [1]. Novel mRNA and replication-deficient vector vaccines can also be administered, as discussed later in the SARS-CoV-2 section.

In major antibody deficiencies (such as CVID, XLA and HIGM) live attenuated vaccines are not recommended [9] but, according to the CDC, they may be considered on the basis of the patient’s risk of exposure and immune status [19]. In general, the measles, mumps, rubella and varicella vaccines should not be given in major antibody deficiencies, as well as *Rotavirus* vaccines [8]. Live attenuated influenza vaccine (LAIV) [4] and oral poliovirus (OPV) administration is not recommended in patients with PAD and their relatives. Indeed, the occurrence of CNS infection after poliovirus vaccine has been reported in XLA and CVID patients [20,21] in particular, when not receiving gamma globulin replacement [22]. Moreover, PAD patients who receive oral poliovirus (OPV) vaccine may excrete poliovirus for a long time [19]. No data are available on efficacy and safety of yellow fever vaccines in PAD patients who might be at increased risk of adverse events; thus, this vaccine is still contraindicated [23].

BCG [20] and *Salmonella typhi* live attenuated vaccines are also currently contraindicated in major PADs [9], despite a recent study showing no systemic reactions on 50 XLA patients vaccinated with BCG and suggesting its use in these patients [20].

Minor antibody deficiencies include selective IgA deficiency (SIGAD), specific polysaccharide antibody deficiency (SPAD) and isolated IgG subclass deficiency. Those patients should receive vaccinations as scheduled for a healthy population [24], although antibody response may be decreased [20]. Both live and inactivated agents can be used, even if there are some special cases. It is known that IgA deficiency can occasionally progress to CVID, leading to a hypothetical risk with live attenuated vaccination; despite that, varicella and measles/mumps/rubella (MMR) vaccines can be scheduled, since there are no complications reported in the literature [25]. On the contrary, OPV is not recommended in these patients and in their close contacts; the indication of BCG vaccine is also debated [7,9]. In addition, conjugate vaccines in minor antibody deficiency may require repeated doses to provide protection [9].

Annual vaccination against influenza virus, in particular, is recommended in PAD patients and, also in household contacts, due to the annual variability of the viral antigens [4].

In the case of international travel, specifically required vaccinations should be individually evaluated. Rabies vaccine is an inactivated vaccine, safe in immunocompromised patients for pre-exposure prophylaxis in high-risk occupations and post-exposure to an infected animal, together with rabies immune globulin or when travelling to endemic areas. Yellow fever vaccine is a live attenuated vaccine and should not be administered to patients with severe humoral defects. It may be considered, in patients with minor antibody deficiencies, according to the risk–benefit balance. Inactivated Japanese encephalitis vaccine is expected to be safe, as are other inactivated vaccines [9].

Pregnant women with PAD should follow the current vaccine recommendations regarding pregnancy and receive TDP and Influenza vaccinations, as these are safe in PAD patients [26,27].

According to the suggestions of the Medical Advisory Committee of the Immune Deficiency Foundation [20] and of the Italian Primary Immunodeficiency Network (IPINET) centers [9], family members and caregivers of patients with PADs should be vaccinated with all available vaccines (except for live poliovirus and live influenza virus).

An antibody defect may be also present in combined and syndromic immunodeficiencies. However, in these patients the vaccine schedule depends on the impairment of T cell function [10].

Table 1 recapitulates the vaccination schedule in PAD patients, according to recent guidelines and recommendations [9].

In secondary antibody deficiencies the vaccination schedule is influenced by the underlying condition (e.g., lymphoproliferative disease), that may be active or in remission, and by the eventually undergoing treatment (e.g., chemotherapy, anti-CD20, etc.). Vaccines containing purified antigens or inactivated organisms, including pneumococcal or influenza vaccine, are generally safe and not associated with increased risk of adverse events. Live vaccines are instead contraindicated or should only be administered after evaluation of the risk–benefit balance, during disease remission and after stopping chemo-immunotherapy [9]. The assessment of the response to specific antigens, such as PPSV23 serotypes, is also used to define the indication to IGRT [28].

## 5. Vaccination and IgRT

For patients receiving IGRT, it is important to know when the last Igs administration was performed. Administration of IGRT should be separated from any other new drug administration, whenever possible, to facilitate the interpretation of any possible adverse reaction or symptom occurring in the hours or days following. Available Igs preparations contain a wide range of specific antibodies and may exert different antimicrobial effects. There is not a plausible rationale for why intravenous Ig (IVIg) should reduce the effectiveness of inactivated vaccines [4]. Consequently, inactivated vaccines (as influenza vaccine) may be administered at any time. On the contrary, IGRT may influence the response to measles, rubella and varicella live vaccines. The effect on mumps vaccine is not known. Thus, it has been recommended to avoid administering these vaccines for 3 to 11 months after the last immunoglobulin administration [4,7]. Hepatitis B, tetanus, rabies postexposure prophylaxis should be carried out with simultaneous administration of immunoglobulin and vaccine [10].

## 6. PAD and COVID-19 Immunization

Since February 2020 we have been experiencing a pandemic caused by a new coronavirus (Sars-CoV-2), causing COVID-19 disease, that has led to over two million deaths worldwide to date [29].

Social distancing, the use of masks and frequent hand washing have become the cornerstone of the fight against COVID-19 disease. At present COVID-19 behavior in patients with inborn errors of immunity (IEI) seems not to differ from the general population, except for a younger age and a longer-lasting SARS-CoV-2 swab positivity. Most patients with antibody production defects do not experience severe disease, even though co-morbidity seems to be crucial in determining outcomes [30,31].

Since the beginning of the COVID-19 epidemic in Italy, most PAD patients have been informed about safety measures and shifted to home therapy and to remote assistance, in order to reduce the risk of infection [32]. At present, the treatment with immunoglobulins does not provide immunity against Sars-CoV-2 infection. For this reason, the European Society for Immunodeficiencies (ESID) has promulgated a statement, shared with the main immunological societies and constantly updated, in which all COVID-19 vaccinations that are not live vaccines are recommended in patients with primary immunodeficiencies, including PADs. The rationale, as for the flu vaccine, is that T cell responses can be generated even in the absence of a complete antibody response [33]. Recent evidence suggests that CVID patients may develop an antibody response and in vitro T-cell reactivity to SARS-CoV-2 antigens [34].

Starting from December 2020, the EMA approved Pfizer/Biontech [35] and Moderna [36] mRNA-based vaccines to fight against COVID-19 infection, followed by the AstraZeneca and Johnson & Johnson adenoviral vector-based vaccines [37]. Others are waiting on approval. These vaccines [38] are able to induce strong CD8^+^ T cell responses, due to the presentation of endogenously produced antigens on MHC class I molecules, in addition to potent CD4^+^ T cell responses and the generation of potent neutralizing antibodies.

Currently, no specific data are available on the efficacy and safety of COVID-19 vaccinations in patients with primary immunodeficiencies. However, in line with what is recommended for other inactivated vaccines, the ESID advises [33] that PAD patients have to be vaccinated according to their national vaccine recommendations. In a recent position statement, the European Board & College of Obstetrics and Gynaecology (EBCOG), despite acknowledging that there is limited evidence on the long-term safety, suggests that the possibility of COVID-19 vaccination should be offered to all pregnant women, after being adequately informed of the benefits and risks. EBCOG also supports that COVID-19 vaccination be recommended to all breastfeeding women, in the absence of a specific contraindication. Due to what has previously been discussed, there is no reason why these suggestions should not be extended to PAD pregnant women [39].

## 7. Conclusions

Since Jenner tested the smallpox vaccine in 1798, millions of lives have been saved. Vaccinations are a safe and effective tool preventing infectious diseases, not only by their direct effect on vaccinated subjects, but also indirectly, providing protection to unvaccinated people (the so-called herd immunity). Vaccines are also effective in the prevention of tumors, (HPV and HBV). Moreover, vaccines play an important role in PAD because they are useful for the diagnosis of antibody deficiencies while still being protective in patients with PADs. Considering that pneumococcal vaccination is increasingly entering into vaccination schedules, and due to the aforementioned evidence, S. typhi Vi vaccine may also be considered in the diagnostic work-up of PAD patients’ diagnosis. Moreover, IgA response to PPSV vaccine could be used as a prognostic marker in CVID patients. A vaccination schedule is currently available for patients with PAD: from this schedule live vaccines are excluded only in severe PADs. Unfortunately, no data are yet available on the efficacy of anti-COVID-19 vaccines in this group of patients, although they were included in the vaccination campaign.

## Figures and Tables

**Table 1 vaccines-09-00626-t001:** Recommendations for Vaccinations in Antibody Deficiency Disorders [7,9,10].

	XLA	CVID	HIGM syn	Thymoma with Immuno Deficiency	SPAD	Isolated IgG Deficiency	SIGAD	IgG Subclass Deficiency	IgA with IgG Subclass Deficiency	Selective IgM Deficiency	THI	Unclassified Antibody Deficiency
**TDP**	Yes	Yes	Yes	Yes	Yes	Yes	Yes	Yes	Yes	Yes	Yes	Yes
**HBV**	Yes	Yes	Yes	Yes	Yes	Yes	Yes	Yes	Yes	Yes	Yes	Yes
**IPV**	Yes	Yes	Yes	Yes	Yes	Yes	Yes	Yes	Yes	Yes	Yes	Yes
**Hib**	Yes	Yes	Yes	Yes	Yes	Yes	Yes	Yes	Yes	Yes	Yes	Yes
**Rotavirus**	No	No	No	No	Yes	Yes	Yes	Yes	Yes	Yes	Yes	NDA
**Pneumo**	Yes	Yes	Yes	Yes	Yes	Yes	Yes	Yes	Yes	Yes	Yes	Yes
**Meningo**	Yes	Yes	Yes	Yes	Yes	Yes	Yes	Yes	Yes	Yes	Yes	Yes
**MMRV**	No	No	No	No	Yes	Yes	Yes	Yes	Yes	Yes	Yes	NDA
**Influenza**	Yes	Yes	Yes	Yes	Yes	Yes	Yes	Yes	Yes	Yes	Yes	Yes
**HPV**	Yes	Yes	Yes	Yes	Yes	Yes	Yes	Yes	Yes	Yes	Yes	Yes
**BCG**	No	No	No	No	Yes	Yes	Yes	Yes	Yes	Yes	Yes	NDA
**S. typhi**	No	No	No	No	Yes	Yes	Yes	Yes	Yes	Yes	Yes	NDA

XLA: x-linked agammaglobulinemia; CVID: common variable immunodeficiency; HIGM Syn: hyper IgM syndrome; SPAD: specific polysaccharide antibody deficiency; SIGAD: selective IgA deficiency; THI: transient hypogammaglobulinemia of infancy. NDA: no data available. TDP: tetanus–diphtheria–pertussis vaccine; HBV: hepatitis B virus vaccine; IPV: inactivated polio vaccine; Hib: *Haemophilus influenzae* B vaccine; Pneumo: pneumococcus vaccine; Meningo: meningococcus vaccine; MMRV: measles–mumps–varicella vaccine; HPV: human papillomavirus vaccine; BCG: bacillus Calmette-Guérin: vaccine used against *Mycobacterium tubercolosis*; S. typhi: *Salmonella typhi* attenuated Ty21a vaccine.

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
