# Peer review of "Vaccination in PADs"

_vaccines, 2021, doi:10.3390/vaccines9060626_

Round 1
Reviewer 1 Report
Review for Vaccination in PADs
This review adds to the literature in that it focuses on vaccine recommendations in primary antibody deficiencies, while most similar reviews focus on primary immunodeficiencies more broadly. Because the authors focused on antibody deficiencies, they are able to provide guidance on more rarely encountered and specific deficiencies and give recommendations. The authors highlight some of their interesting work on the utility of IgA in prognosticating for this group of patients. Overall, the paper would benefit from some editorial review.
General
- Please do not capitalize the second name for species names.
- Please be consistent in your abbreviation for IGRT.
Abstract
- Since this will be read by an international worldwide audience, please spell out EMA.
- The sentence beginning on line 19 is unclear. Perhaps stating vaccination may be unsafe would assist in providing clarity.
- I would consider adding a statement regarding novel strategies regarding diagnosis and prognosis in this population (referring to the sections on the use of Typhim Vi vaccine and IgA and IgM discussed in the text).
Introduction
- Please spell out correct name fully for IUIS.
- For the sentence beginning on line 54, do you have a reference regarding HPV titers in Ig preparation? I would add the words “currently circulating” prior to influenza to more accurately reflect what is reported in the included reference.
Rationale of vaccination in PAD
- For the first sentence, controversial might be a better word than critical.
- In the second paragraph, did you find any evidence of an association between vaccination in this population and improved outcomes related to vaccine-preventable disease (beyond evidence of immune response)?
- The first sentence in the paragraph beginning on line 82 is a run-on sentence. Please break into smaller sentences to improve readability.
- The sentence beginning, “of note, …” that starts on line 90 is awkward. I think the point you are making is that already high antibody titers are less likely to increase as much as lower initial titers after immunization, suggesting that a threshold of a 2-fold increase would be more widely applicable. I would reword the sentence to make this more clear if this is in fact what you are trying to convey.
- In the paragraph beginning, “Wall et al…”, please specify what is considered the protective range mentioned on line 96. Also what is meant by the percentage response? Is this the percentage of tested serotype titers meeting the threshold of response? Once the percentage response is determined, how should that be interpreted? How does this information change management of the patient?
- On line 102, please define CDC.
- I’m not sure the sentence beginning on line 105 is relevant to the discussion. I would consider removing.
Prognostic role of vaccination
- In the second paragraph beginning on line 132, please explicitly state the number of patients divided into the 3 groups.
- How long after vaccination were these measurements taken?
- In the third paragraph, were the described patients PCV naïve?
- The sentence beginning on line 142, I think the word “patients” is missing, which would improve the clarity of the sentence.
- At what point was the 3rd evaluation described on line 148?
- Were the comparisons described on lines 148-150 statistically significantly different?
Vaccination schedule in PAD and recommendations
- On line 169, please specify oral poliovirus vaccine.
- Is rotavirus administration recommended in the relatives of patients with PAD?
- On line 178, is reference 18 appropriate here? Is reference 20 the one describing the study mentioned?
- Can you given an example of such a special case, mentioned on line 184?
Table 1.
- I would consider changing the title to “Recommendations for Vaccinations in Antibody Deficiency Disorders.”
- Consider adding the reference for Table 1 into the legend.
- In the legend, please specify which Salmonella typhi vaccine. (Typhi is misspelled in the legend.)
PAD and COVID-19 immunization
- Please be consistent with COVID-19 abbreviation.
- In the sentence beginning on line 241, what are you saying occurs at a younger age? Symptoms? Infection? Increased severity? Please specify.
- In the sentence beginning on line 243, which comorbidities are crucial in determining outcome in this population?
- The sentence beginning on line 262 is unclear. Perhaps it should refer to eligibility criteria for COVID-19 vaccines rather than the national vaccination schedule, since I don’t think this vaccine has been incorporated into most countries’ routine schedule thus far.
Conclusions
- I would spend less time here talking about the history of vaccines in general and focus more on what the focus has been in this paper, such as that vaccines are still protective in patients with PAD.
- You could consider mentioning utility of Typhim Vi vaccine here as this could represent a change in practice similar to that described with use of IgA.
- Can also mention here that generally live vaccines are contraindicated in severe PAD.
Author Response
Review for Vaccination in PADs
This review adds to the literature in that it focuses on vaccine recommendations in primary antibody deficiencies, while most similar reviews focus on primary immunodeficiencies more broadly. Because the authors focused on antibody deficiencies, they are able to provide guidance on more rarely encountered and specific deficiencies and give recommendations. The authors highlight some of their interesting work on the utility of IgA in prognosticating for this group of patients. Overall, the paper would benefit from some editorial review.
General
- Please do not capitalize the second name for species names. Thank you for your suggestion.
- Please be consistent in your abbreviation for IGRT. Thank you for your suggestion
Abstract
- Since this will be read by an international worldwide audience, please spell out EMA. Thank you for your suggestions
- The sentence beginning on line 19 is unclear. Perhaps stating vaccination may be unsafe would assist in providing clarity. We modified according to your suggestion.
- I would consider adding a statement regarding novel strategies regarding diagnosis and prognosis in this population (referring to the sections on the use of Typhim Vi vaccine and IgA and IgM discussed in the text). Thank you. We modified according to your suggestion.
Introduction
- Please spell out correct name fully for IUIS. Thank you for your suggestions. We modified.
- For the sentence beginning on line 54, do you have a reference regarding HPV titers in Ig preparation? We contacted several Companies, but, at now no data are available on HPV titers in Ig preparation .
- I would add the words “currently circulating” prior to influenza to more accurately reflect what is reported in the included reference. Thank you for your suggestions. We added.
Rationale of vaccination in PAD
- For the first sentence, controversial might be a better word than critical. Thank you. We modified.
- In the second paragraph, did you find any evidence of an association between vaccination in this population and improved outcomes related to vaccine-preventable disease (beyond evidence of immune response)? We modified sentence according to your suggestion.
- The first sentence in the paragraph beginning on line 82 is a run-on sentence. Please break into smaller sentences to improve readability. Thank you. We modified.
- The sentence beginning, “of note, …” that starts on line 90 is awkward. I think the point you are making is that already high antibody titers are less likely to increase as much as lower initial titers after immunization, suggesting that a threshold of a 2-fold increase would be more widely applicable. I would reword the sentence to make this more clear if this is in fact what you are trying to convey. Thank you. We tried to make the sentence more clear.
- In the paragraph beginning, “Wall et al…”, please specify what is considered the protective range mentioned on line 96. Also what is meant by the percentage response? Is this the percentage of tested serotype titers meeting the threshold of response? Once the percentage response is determined, how should that be interpreted? How does this information change management of the patient? Thank you.
In the manuscript we underlined Dall’articolo “When gauging a sufficient PPV23 response for an individual patient, one approach in the past has been to compare postimmunization serotype-specific antibody concentrations with preimmunization concentrations, considering normal to be a 4-fold increase in the concentration. Most experts now agree that the most reliable and straightforward approach in gauging response to PPV23 is to consider the percentage of serotype-specific titers measured (titers to serotypes included in PPV23) that are within the protective range post-PPV23. An acceptable percentage of protective serotypes is greater than or equal to 50% of serotypes for patients less than 6 years of age, and greater than or equal to 70% of serotypes of patients aged greater than or equal to 6 years.”
- On line 102, please define CDC. We defined it.
- I’m not sure the sentence beginning on line 105 is relevant to the discussion. I would consider removing. Thank you. We decided to remove it.
Prognostic role of vaccination
- In the second paragraph beginning on line 132, please explicitly state the number of patients divided into the 3 groups. As requested, we stated the number of patients for each group.
- How long after vaccination were these measurements taken? As already reported in line 130: “we evaluated pre and 4 weeks after immunization with PPSV23 vaccine, by using an ELISA test to detect IgA and IgM antibodies produced against all 23 PS serotypes”.
- In the third paragraph, were the described patients PCV naïve? Yes, they were PCV naive. We decided to test IgA and IgM because of they were on IgG replacement therapy.
- The sentence beginning on line 142, I think the word “patients” is missing, which would improve the clarity of the sentence. Thank you for your suggestions. We added the word “patients” to improve the clarity of the sentence.
- At what point was the 3rd evaluation described on line 148? As reported on line 146, the 3rd evaluation was performed 36 ± 6 months after the first immunization
- Were the comparisons described on lines 148-150 statistically significantly different? We observed statistically significant differences between group and according to this, we modified sentence.
Vaccination schedule in PAD and recommendations
- On line 169, please specify oral poliovirus vaccine. Thank you for your suggestion. We specified.
- Is rotavirus administration recommended in the relatives of patients with PAD? Rotavirus can be administered to family members or other close contacts susceptible to infection, since the risk of developing the disease is extremely rare.
- On line 178, is reference 18 appropriate here? It was a mistake. The appropriate reference is 9.
- Is reference 20 the one describing the study mentioned? Yes, reference 20 describes the study on XLA patients.
- Can you given an example of such a special case, mentioned on line 184? A patient affected by selective IgA deficiency is a special case.
Table 1.
- I would consider changing the title to “Recommendations for Vaccinations in Antibody Deficiency Disorders.” Thank you. We modified title according to your suggestion.
- Consider adding the reference for Table 1 into the legend. We added references 7, 9 and 10.
- In the legend, please specify which Salmonella typhi vaccine. (Typhi is misspelled in the legend.) Thank you. We specified.
PAD and COVID-19 immunization
- Please be consistent with COVID-19 abbreviation. Thank you. We modified.
- In the sentence beginning on line 241, what are you saying occurs at a younger age? Symptoms? Infection? Increased severity? Please specify. We mean that, differently from general population, IEI patients affected by COVID-19 requiring Intensive Care Unit (ICU) admission were younger. Moreover, as described in Ref. 28 “despite several comorbidities, most of them present a milder course of the disease or are even asymptomatic. “Differently from the general population, patients with IEI and severe COVID-19.
- In the sentence beginning on line 243, which comorbidities are crucial in determining outcome in this population? All adult patients with IEI who died had comorbidities (chronic lung/heart disease, obesity, diabetes, hypertension) whereas pre-existing heart, lung, or kidney disease are risk factors for severe COVID-19 in patients with IEI seem very similar to those in the general population.
- The sentence beginning on line 262 is unclear. Perhaps it should refer to eligibility criteria for COVID-19 vaccines rather than the national vaccination schedule, since I don’t think this vaccine has been incorporated into most countries’ routine schedule thus far. At now, COVID-19 vaccines are not incorporated in routine schedule but, for fragile patients, are recommended by Italian Health Ministry. We modified “schedule” with “recommendation”.
Conclusions
- I would spend less time here talking about the history of vaccines in general and focus more on what the focus has been in this paper, such as that vaccines are still protective in patients with PAD. Thank you. We modified according to your suggestion.
- You could consider mentioning utility of Typhim Vi vaccine here as this could represent a change in practice similar to that described with use of IgA. Thank you. We modified according to your suggestion.
- Can also mention here that generally live vaccines are contraindicated in severe PAD. Thank you. We modified according to your suggestion.
Reviewer 2 Report
The Review by Milito at al. deals with an important issue of vaccination schedule for patients affected with Immunodeficiencies. The paper deserves full consideration, even though a few points need to be addressed to improve the strength of the paper:
- The Authors should mention that detailed information on the duration of the protection following vaccination in PAD is missing. Additional boosters over time may be required;
- The Authors should mention/propose a strategy to follow in those cases with a different response to different vaccines (e.g. good response to polysaccharide ag, and absent response to protein ag, as HBsAg);
- Although the focus is on primary IDs, a short comparison wth acquired ADs may help the reader. A mention on the significance of qualitative vs quantitative defects may be useful.
Author Response
The Review by Milito at al. deals with an important issue of vaccination schedule for patients affected with Immunodeficiencies. The paper deserves full consideration, even though a few points need to be addressed to improve the strength of the paper:
- The Authors should mention that detailed information on the duration of the protection following vaccination in PAD is missing. Additional boosters over time may be required. Thank you for your suggestion, we included it within the manuscript, line 162-163.
- The Authors should mention/propose a strategy to follow in those cases with a different response to different vaccines (e.g. good response to polysaccharide ag, and absent response to protein ag, as HBsAg). Author position: in case of uneven response to T-dependent and T-independent vaccines, is to consider those patients as non-responder.
- Although the focus is on primary IDs, a short comparison with acquired ADs may help the reader. A mention on the significance of qualitative vs quantitative defects may be useful. Even though secondary antibody deficiency is not the target of the present manuscript, according to reviewer 2 suggestion, we make some consideration regarding this topic, lines 211-219.
Reviewer 3 Report
A simple but efficient description of the use of vaccination in patients with antibody deficiency is presented. It is objective and transmits important basic knowledge.
Made of minus several aspects:
1. Should the vaccine be separated from the administration of immunoglobulins?
2. Vaccination of pregnant women with antibody deficiency, how to manage the vaccine in the newborn?
3. Vaccination of families and partners of patients with antibody immunodeficiency, to whom and when?
Author Response
A simple but efficient description of the use of vaccination in patients with antibody deficiency is presented. It is objective and transmits important basic knowledge.
Made of minus several aspects:
Should the vaccine be separated from the administration of immunoglobulins? In point 5, we discuss “Vaccination and IgRT” and, according to your suggestion, we underline this issue adding the following sentence “Administration of IGRT should be separated from any other new drugs administration, whenever possible, to facilitate the interpretation of any possible adverse reaction or symptom occurring in the hours or days following”.
2. Vaccination of pregnant women with antibody deficiency, how to manage the vaccine in the newborn? Thank you for the suggestion, we included it within the manuscript, line 207-209. “Pregnant women with PAD should follow the current vaccines recommendation regarding pregnancy and receive TDP and Influenza vaccination, as there are safe in PAD patients”.- Vaccination of families and partners of patients with antibody immunodeficiency, to whom and when? As already reported in the text, point 4, line 207-210: “According to the suggestions of the Medical Advisory Committee of the Immune Deficiency Foundation [20] and of the Italian Primary Immunodeficiency Network (IPINET) Centers [9], family members and caregivers of patients with PAD should be vaccinated with all available vaccines (except for live poliovirus and live influenza virus).”
